# A Workplace Health Promotion Program for a Predominantly Military Population: Associations with General Health, Mental Well-Being and Sustainable Employability

**DOI:** 10.3390/ijerph21050625

**Published:** 2024-05-15

**Authors:** Rebecca Bogaers, Diewertje Sluik, Pieter Helmhout, Fenna Leijten

**Affiliations:** 1Strategic Military Healthcare Department, Defence Healthcare Organisation, Ministry of Defence, 3584 AB Utrecht, The Netherlands; d.sluik@mindef.nl (D.S.); 2Centre of Excellence, Training Medicine and Training Physiology, Royal Netherlands Army, Ministry of Defence, 3584 AB Utrecht, The Netherlands; 3Directorate of Strategy and Knowledge, Directorate-General of Policy, Ministry of Defence, 2511 CB The Hague, The Netherlands

**Keywords:** workplace health programs, health, sustainable employability, military

## Abstract

Due to the globally increasing life expectancies, many countries are raising their official retirement age to prevent labor shortages and sustain retirement systems. This trend emphasizes the need for sustainable employability. Unhealthy lifestyles pose a risk to sustainable employability as they contribute to chronic diseases and decreased productivity. Workplace Health Promotion (WHP) programs have gained attention as a strategy to enhance employee health and well-being. The Netherlands Armed Forces, a unique employer with demanding psychological and physical requirements, was used as a case study to investigate the associations of a WHP Program with workers health and sustainable employability. The program offered tailor-made guidance to participants (*N* = 341) through individual coaching trajectories. The program’s impact was evaluated by measuring self-reported health, mental well-being, and sustainable employability over a 6-month period. Results indicated significant improvements across all these dimensions after participation in the program. This study provides valuable insights into the benefits of tailor-made WHP programs. While this was an observational study without a control group, this study supports the importance of incorporating individualized approaches in WHP initiatives to foster positive outcomes in health and sustainable employability.

## 1. Introduction

Due to medical, economic, and social developments, life expectancies are increasing worldwide [1]. Consequently, there is a higher financial burden on the working population to take care of the retired, older adults and the sick. To prevent labor market shortages and to sustain retirement systems, many countries have decided to raise their official retirement age [2]. As employees have to work longer, the importance of sustainable employability has increasingly gained attention [3].

One of the risk factors for (chronic) diseases and reduced sustainable employability in working populations, in terms of lost working days and reduced productivity, is an unhealthy lifestyle, including low physical activity and a poor diet [4,5]. Conversely, health promotion can result in improved sustainable employability. For example, workplace nutrition and physical activity interventions have shown to improve productivity, work performance, and work ability [6]. Therefore, it is important for employers, and society at large, to invest in employee health.

The workplace is an important environment for health promotion, due to its opportunity to reach a large group of people within a social network structure. Thus, in order to maintain or improve employees’ health and work performance, employers may introduce Workplace Health Promotion (WHP) using educational and/or intervention programs [7]. The goal of WHP is to positively influence the lifestyle of the workforce and to improve their health, work productivity, and work ability [5]. However, research on the effects of WHP is limited, and the findings are non-conclusive. There are no standards for WHP programs that have been reported in the literature: definitions, activities, execution, and outcomes vary drastically. As a result, they are difficult to compare. However, several studies have shown positive effects of WHP on employee health, well-being, and productivity [6,8,9], and the current study aims to contribute to the evidence base of WHP.

The Netherlands Armed Forces (NAF), with approx. 68,000 employees, one of the largest organizations in the Netherlands, is an integrated workforce with military and civilian personnel. Particularly for military service members, sustainable health and readiness are important, given the high psychological and physical demands that are required for their duties. 

The current study focuses on a WHP program, the so-called "Health and Readiness" Program, that was implemented within the NAF. This Health and Readiness Program was based on the multifaceted definition of ‘positive health’. This definition goes beyond the absence of disease or infirmity and encompasses various dimensions of health, emphasizing the promotion of physical, mental, emotional, and social well-being [10]. A holistic approach was used, considering various aspects of a person’s life, such as their physical health, mental well-being, and personal values. Furthermore, the Health and Readiness Program consisted of tailor-made modules and was executed by a professional (lifestyle) coach. Tailor-made programs are essential when targeting health and sustainable employability, as they recognize and address the unique individual needs of each person. Research shows that tailor-made programs can enhance effects, participation, and compliance [11,12]. Furthermore, the use of personal coaching by a professional coach was shown to be important for the effectiveness of WHP [12,13]. Finally, one of the goals of the current WHP Program was to provide employees with insight into their current health behaviors, as this is a crucial component of many behavior change models and theories [14,15]. Understanding one’s behavior, motivations, triggers, and barriers is essential for successfully making positive changes. Together this makes the current WHP Program innovative as it is tailor-made, uses personal coaching, and has a holistic approach. A meta-analysis shows that only a few studies have combined these elements [5]. 

The main goal of the WHP Program was to improve lifestyle, health, and sustainable employability of the employees in several pilot departments of the NAF, by giving them insight into their current health behaviors and providing them with tailor-made guidance to make healthier choices. The objective of this study was to examine associations between participating in the WPH Program and health and sustainable employability improvements over six months and to provide insight into the WHP Program and its strengths and limitations. 

## 2. Materials and Methods

### 2.1. Study Design and Population

This study used a non-randomized one-group pre-test and post-test design. It was carried out within three different NAF Staff departments including military and civilian personnel: the Royal Netherlands Army Staff (November 2018–November 2020) (*N* = 603), the Central Staff (March 2019–March 2021) (*N* = 1430), and the Royal Netherlands Airforce Staff (October 2019–April 2021) (*N* = 472). These departments were selected because they are generally characterized by older personnel with sedentary desk work activities, often exposed to high levels of stress due to a high workload, large responsibilities, and external deadlines. In addition, military staff personnel usually lack the unit-wise physical activity during working hours that is common in operational units. Everyone in the specific departments was allowed to participate, and there were no eligibility criteria. 

All participants provided written informed consent prior to the start of the study. Participation in the WHP Program as well as in the accompanying evaluation with online questionnaires was voluntary: participants could withdraw at any time. The study design was approved by the NAF institutional review board. This study was not subject to the Dutch Medical Research Involving Human Subjects Act.

#### 2.1.1. The Workplace Health Promotion Program 

The Health and Readiness Program was tailor-made. A professional lifestyle coach supported participants in an individual coach trajectory based on their needs and goals. Lifestyle coaches are trained at a higher vocational education level in using coaching skills to help individuals make healthy choices in their daily lives. Two lifestyle coaches were allocated to the Airforce Staff and three lifestyle coaches each to the Army Staff and the Central Staff. All employees working at the respective departments could voluntarily sign up. The program will be described in more detail below. An overview of the process of the WHP Program can be found in Figure 1. 

##### Kick-Off

Each program started with a kick-off event for all department employees, after which they could sign up to participate. The purpose of this event was to draw attention to the program and its importance to senior leadership. The event consisted of information stands, an inspirational speaker, and catering. 

##### Baseline Questionnaire

After the kick-off event, everyone in the different participating military departments received an e-mail in which information was provided about what participation consisted of. Those who wanted to participate, received a link to fill out the online baseline questionnaire (M0), followed by an intake meeting with one of the lifestyle coaches (hereafter, coach).

##### Intake Meeting

During the first intake meeting with the coach, several topics were discussed such as the results of the baseline questionnaire, background information of the participant, whether they were receiving other health care, the mutual expectations, and their motivation for participation. Following this, goals were set based on the individual needs of the participant.

##### Coach Trajectory

The participant subsequently met periodically (typically every 2–4 weeks) with the coach for guidance and to discuss progress. The intensity and duration of the coach trajectory was based on participants’ individual needs. On average, participants met with the coach 4–9 times. Initially, the meetings took place face-to-face. However, as COVID-19 regulations were initiated in March 2020 and people were asked to work from home, meetings also took place online. Research suggests that online coaching is equally effective as in-person methods [16]. 

The coach kept track of every meeting by taking notes about the topics of the meeting, the progress of the participant, and aspects that stood out to the coach. The participants were asked to reflect on each meeting and to write a short report. The topics of the report covered insights gained during the meeting, the activities the participant would undertake to reach their goal, and the activities they would stop doing. They e-mailed this short report to the coach and the coach used this report as input for the next meeting. 

Although all coaches had their own methods of working, motivational coaching was commonly used [17]. The coaches also had several additional instruments that could be used to support the participants. For example, they had tools to assess mental health, the possibility to weigh participants, fitness trackers, and the possibility of administering the dietary behaviors of participants. Also, there was the possibility of administering a health check, where participants’ blood-pressure and cholesterol levels were checked. Finally, the coaches were encouraged to attend multidisciplinary professional consultation meetings within their staff departments. 

##### The 6-Month Questionnaire

All participants were asked to fill out an online questionnaire, similar to baseline, 6 months after their baseline measurement (M1). At M1, some coaches had already finalized their coach trajectory, and some were still on the coach trajectory. 

### 2.2. Measurements

Participants received an invitation by e-mail to fill out an online questionnaire at baseline and 6 months after the baseline measure. The tool ‘LimeSurvey voor Defensie: Versie 1.91+ Build 10746’ was used, the internal survey tool of the NAF. The questionnaire assessed demographics (only at baseline), the goals that were set (only at baseline), whether goals were reached (only after 6 months), general health and body mass index (BMI; kg/m^2^), mental well-being (work engagement, burn-out complains) and indicators of sustainable employability (recovery, work–home balance, work ability).

#### 2.2.1. Background 

Demographics: Age category (<35, 35–50, >50), sex (male, female), educational level (low, middle, high), rank (non-commissioned officer, subaltern officer, chief officer or higher), pay grade (low, middle, high), and form of employment (military, civilian, reserves, other) were measured.Goals set: Participants were given a range of goals and were asked to indicate for each goal whether this was a goal they had set for themselves (e.g., lose weight, improve diet, improve mental health).Goals reached: Participants were asked whether they had reached the goals that they had set for themselves, with the following answer categories: ‘completely reached goals that were set’, ‘almost reached the goals that were set’, ‘did not reach the goals that were set’, or ‘no goals were set’.

#### 2.2.2. Health

General health: The first item of the Short-Form Health Survey 12 (SF-12) “What do you think, in general, of your health?” was used, measured on a 5-point Likert scale ranging from 1 = poor to 5 = excellent, with higher scores indicating better general health [18].BMI: This was calculated based on self-reported height and weight (kg/m^2^).

#### 2.2.3. Mental Well-Being

Work engagement: A mean score of three items from the Utrecht Work Engagement Scale (UWES) was used. The items were measured on a five-point Likert scale ranging from 1 = never to 5 = always with higher scores indicating higher work engagement [19].Burn-out complaints: A mean score of four items from the Utrecht Burnout Scale (UBOS) was used. The items were measured on a five-point Likert scale ranging from 1 = never to 5 = always with higher scores indicating more burn-out complaints [20].

#### 2.2.4. Sustainable Employability

Recovery: As an indication of the ability to recover after work, a mean score of three items from the Demand-Induced Strain Recovery Questionnaire was used. The items were measured on a five-point Likert scale ranging from 1 = never to 5 = always, with higher scores indicating better ability to recover after work [21].Work-home balance: One item was used, namely “My work situation and home situation combine well”. This item was measured on a five-point Likert scale ranging from 1 = never to 5 = always with higher scores indicating a better work–home balance.Work ability: Work ability was defined as the degree to which employees can meet the demands of their work both physically and mentally. It was assessed with the first item of the Work Ability Index: “If during the best period of your life, your work ability would be graded a 10, how would you grade your current work ability?”, with scores ranging from 1 to 10 with higher scores indicating better work ability [22].

### 2.3. Statistical Analyses

Questionnaires were anonymized. Researchers who performed the data analyses only had access to the anonymous data, stored on a secured drive. Statistical analyses were performed using IBM SPSS Statistics 25, IBM Corporation, New York, NY, USA. Descriptive analyses were used for the sample characteristics and intervention characteristics. Only those who completed M0 and M1 were included for further analyses. To assess 6-month differences, Wilcoxon Sign-Rank tests were used, as the outcome variables were not normally distributed. Additionally, to examine the robustness of the results, linear regression analyses were used to determine whether the 6-month differences depended on sex, age, or educational level. Furthermore, a one-way ANOVA test was used to test for differences between the participating military departments in baseline and 6-month scores of health, mental well-being, and sustainable employability. A significance level of *p* < 0.05 was used. 

## 3. Results

### 3.1. Response Rate

Figure 2 displays the response rates per department. Within the Army Staff (*N* = 603), *N* = 261 (43.3%) filled out the baseline questionnaire (M0), of whom *N* = 159 (60.9%) also filled out the 6-month questionnaire (M1). Within the Central Staff (*N* = 1430), *N* = 227 (15.9%) filled out the baseline questionnaire (M0), of whom *N* = 136 (59.9%) also filled out the 6-month questionnaire (M1). Within the Airforce Staff (*N* = 472), *N* = 154 (32.6%) filled out the baseline questionnaire (M0), of whom *N* = 46 (29.9%) also filled out the 6-month questionnaire (M1). It should be noted that the number of employees participating in the coaching trajectory was also influenced by the maximum workload of the coaches. 

### 3.2. Sample Characteristics

The sample characteristics can be found in Table 1. 

### 3.3. Intervention Characteristics

The characteristics of the intervention, such as the number and duration of consultations with the coach, can be found in Table 2. 

### 3.4. Changes in Health, Mental Well-Being, and Sustainable Employability in 6 Months

As can be seen in Table 3, there was a significant change during the 6 months for all the measures of health, mental well-being, and sustainable employability. The largest changes were observed for general health (*M*_M0_ = 3.05, *M*_M1_ = 3.33, *Z* = −7.338, *p* < 0.001), recovery after work (*M*_M0_ = 2.76, *M*_M1_ = 3.06, *Z* = −6.889, *p* < 0.001), BMI (*M*_M0_ = 26.20, *M*_M1_ = 25.75, *Z* = −4.820, *p* < 0.001), and work ability (*M*_M0_ = 7.47, *M*_M1_ = 7.79, *Z* = −4.513, *p* < 0.001). The smallest, yet significant, change was observed for work–home balance (*M*_M0_ = 4.17, *M*_M1_ = 4.27, *Z* = −2.307, *p* < 0.05). 

### 3.5. Robustness of the Results

When comparing the different military departments on their scores of health, mental well-being, and sustainable employability at baseline (M0) and after 6 months (M1), the one-way ANOVA showed no significant differences across the departments except for the baseline measure of work engagement (*F*(2) = 5.285, *p* < 0.05). The Army Staff scored significantly higher on baseline work engagement (*M =* 3.61) compared to the Central Staff (*M =* 3.36) and the Airforce Staff (*M =* 3.38). 

Additional analyses (Appendix A) showed that the 6-month differences presented in Table 3 were not dependent on military department, sex, age, or educational level of participants, except for the difference in work-engagement which was higher for females compared to males. There results indicate that the WPH Program was effective for the different subgroups. 

## 4. Discussion

The current study evaluated a WPH program in the NAF. Significant changes in health, mental well-being, and sustainable employability were observed in the study group. The findings underscore the potential benefits of workplace health promotion programs that use tailor-made guidance and professional coaching. 

Previous research on the effectiveness of WPH programs has yielded inconclusive findings. For example, a randomized clinical trial that examined the association of a comprehensive employee wellness program with health outcomes found no significant associations [23]. However, other studies have found positive effects, with workplace interventions improving productivity, work performance, and work ability [6]. As there are no standards for employee wellness programs in terms of activities, execution, and outcomes, it is difficult to compare the results of different programs.

Pesis-Katz, Norsen, and Singh [24] argue that the effectiveness of WPH programs depends on the incorporation of certain key factors. These key factors include (1) the use of a multidisciplinary model of care that uses a highly skilled team of people certified in wellness coaching; (2) a clinical program that is based on a biopsychosocial framework and includes a sophisticated technological platform to enable individual coaching; (3) clinical integration with the healthcare system; (4) use of techniques beyond financial incentives to improve active participation; and (5) continuous quality improvement supported by the latest evidence. The current WPH Program included most of these key factors. For example, the coaches were encouraged to attend multidisciplinary professional consultation meetings, and they had access to a wide range of tools that could encourage active participation. Additionally, participation in the WPH Program was free of charge and made possible during working hours, possibly making it more attractive to participate. It should be noted that the WPH Program was not clinically integrated within the healthcare system. Though Pesis-Katz, Norsen, and Singh [24] argue that this is important for the success of a WPH program, we believe that one of the success factors of the current WPH Program was that the coach did not represent a specific domain within the healthcare system (e.g., mental/physical/social) and that they, thus, incorporated a holistic approach connected to multiple health (care) professions. Furthermore, the study took place within the military organization, comprising a wide range of available options to encourage a healthy lifestyle. These options could be used by participants on top of their coach trajectory. A broader overview of what the NAF offers in this area can be found in Appendix B. 

The literature shows that strong leadership and leadership support are important for employee well-being and sustainable employability [25,26,27,28]. The current WHP Program included the senior leadership by informing them beforehand about the program. Ultimately, they gave permission for the program and were willing to emphasize the importance of the program at the kick-off event. The involvement of senior leadership is expected to be highly important in military populations, as the strong hierarchical structure in the military makes employees more dependent on their leaders [29]. Implementing WPH programs, especially within the military, should thus always incorporate leadership and educate leaders on how they can support employee well-being and sustainable employability. The current WHP Program could have included senior leadership even more by, for example, asking them to stress the importance of participating in the 6-month questionnaire, which could have decreased the drop-out rates. 

In the current study, coach trajectory participation rates varied per department, with the highest participation rate in the Army Staff (43.3%) and the lowest participation rate in the Central Staff (15.8%). These differences may have been caused by differences in leadership support, a less fruitful match between the department and its allocated coach, or due to the time period in which the program took place and the influence of COVID-19 during this time. However, it is not possible to draw any conclusions in this respect based on the data of the current study. A literature review showed that participation rates in workplace health promotion programs are generally low, especially among those who need them most: rates typically range between 20 and 30% [30], which is comparable to the rates found in the current study (Army Staff 43.3%, Central Staff 15.8%, Airforce Staff 32.6%). The low participation rates could be caused by several reasons. Foremost, it is likely that not everyone in the department had health-related questions that they want to work on, causing them not to participate. Workload may be another reason for the low participation, especially within certain (staff) departments, leaving no time to participate in the WPH Program. Furthermore, there could be stigma related to participating. Previous research has shown that stigma surrounding mental health can form a barrier to help-seeking, especially in the military [31]. For example, military personnel fear that seeking treatment will make others think they are weak, forcing them to solve their own problems [32]. A similar stigma may be associated with participating in WPH programs. Future research should further examine how to take the possible modifying role of stigma into account when developing WPH programs. 

Finally, is important to acknowledge that interventions such as the current WHP Program are limited in their effects, as they do not address work factors that can be a source of poor well-being at work. The interventions focus on how individuals can manage work factors such as workload through healthy lifestyle changes, but they do not focus on decreasing the workload on an organizational level when needed. However, when organizations provide employees with the opportunity to invest in their well-being at work through WHP programs, this gives a signal to employees that their well-being is a priority to the organization. 

### Strengths and Limitations

The current study examined an innovative WPH program using an individual-based approach. With a total of 341 participants who completed the pre- and post-measure, the current study had a large sample size, making our findings more robust.

While the results are promising, several limitations should be acknowledged. First, the lack of a control group limits causal inferences. Future research should explore options to further examine the causal inferences, for example, by including matched control groups or by comparing the results of participants to the results of a general monitor of all employees. A specific confounder that could, for example, explain the positive effects of the program is COVID-19. As people were asked to work from home during the pandemic, this might have improved their work–life balance and burn-out complaints as they saved time commuting. However, for others, it might have increased burn-out complaints as the schools and daycare centers were also closed. Second, the study’s reliance on self-report measures could introduce response biases. However, the self-report measures were only used to assess current health, and there was no recall of previous health, limiting the response bias. Third, the sample is not representative of the whole NAF, with an over-representation of females and an underrepresentation of military service members compared to civilian service members. However, additional analyses revealed that the differences from pre- to post-measure did not depend on the characteristics of the participating group (e.g., sex, education, and age). Additionally, there is a self-selection bias, as participants could freely sign up to participate, meaning that the sample only included motivated people. The results will likely be different for unmotivated participants.

## 5. Conclusions

In conclusion, the results of this study underscore the potential benefits of tailor-made WPH programs for the health, well-being, and sustainable employability of employees. Additionally, the findings contribute to the literature on WPH programs. Further research is warranted to build upon these findings and explore the broader impact of tailor-made WPH programs across diverse work environments. 

## Figures and Tables

**Figure 1 ijerph-21-00625-f001:**
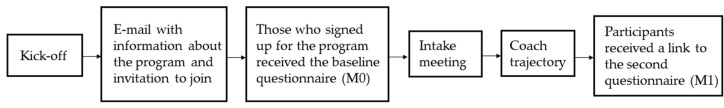
Process of the Workplace Health Promotion Program.

**Figure 2 ijerph-21-00625-f002:**
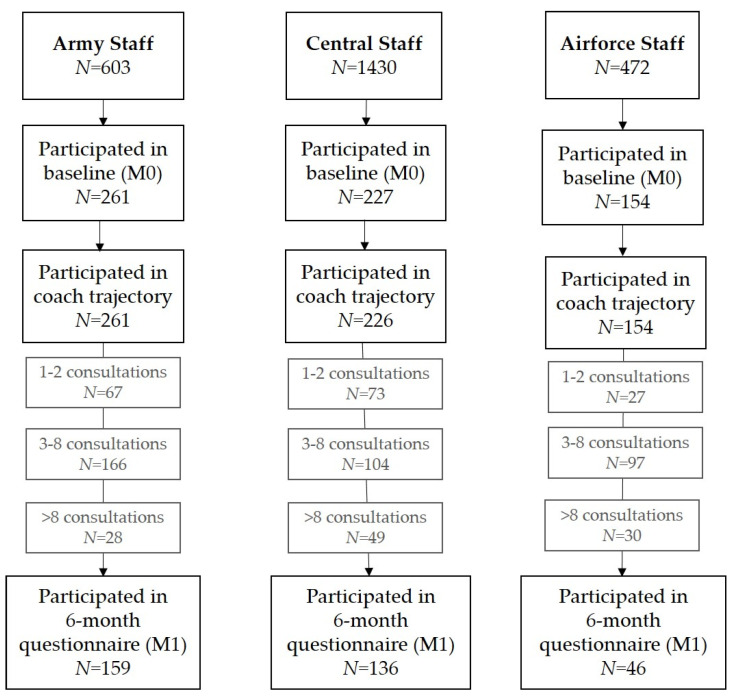
Response rates separated per department.

**Table 1 ijerph-21-00625-t001:** Sample characteristics separated per department and characteristics of the staff departments.

		Total Sample *N* = 341	Army Staff Sample *N* = 159	Army Staff *N* = 603	Central Staff Sample *N* = 136	Central Staff *N* = 1430	Airforce Staff Sample *N* = 46	Airforce Staff *N* = 472
**Sex** **(*N* = 340)**	Male	209 (61.5%)	111 (69.8%)	479 (79%)	75 (55.2%)	998 (70%)	23 (51.1%)	357 (75.6%)
	Female	131 (38.5%)	48 (30.2%)	124 (21%)	61 (44.8%)	432 (30%)	22 (48.9%)	115 (24.4%)
**Age**	<35	42 (12.3%)	15 (9.4%)	56 (9%)	17 (12.5%)	234 (16%)	10 (21.7%)	91 (19.2%)
	35–50	161 (37.2%)	69 (43.4)	227 (38%)	73 (53.7%)	571 (40%)	19 (41.3%)	196 (41.5%)
	>50	138 (40.5%)	75 (47.2%)	320 (53%)	46 (33.8%)	628 (44%)	17 (37.0%)	188 (39.8%)
**Education ***	Low and middle	203 (59.5%)	112 (70.5%)	-	59 (43.4%)	-	32 (69.6%)	-
	High	138 (40.5%)	47 (29.5%)	-	77 (56.6%)	-	14 (30.4%)	-
**Rank** **(*N* = 187)**	Non-commissioned officer	21 (11.2%)	12 (12.4%)	49 (13%)	6 (10.3%)	62 (9%)	-	51 (10.9%)
	Subaltern officer	32 (17.1%)	14 (14.4%)	71 (20%)	8 (13.8%)	85 (13%)	-	25 (5.3%)
	Chief officer or higher	134 (71.7%)	71 (73.2%)	242 (67%)	44 (75.9%)	526 (78%)	19 (59.4%)	299 (79.7%)
**Pay grade** **(*N* = 150)**	Low (1-7)	24 (16%)	17 (27.9%)	53 (22%)	-	70 (9%)	-	21 (21.6%)
	Middle (8-10)	23 (15.3%)	8 (13.1%)	56 (24%)	-	130 (13%)	-	34 (35.1%)
	High (11-18)	103 (68.7%)	36 (59%)	127 (54%)	63 (81.8%)	560 (74%)	-	42 (43.3%)
**Occupation**	Military service member	176 (51.6%)	87 (54.7%)	367 (61%)	57 (41.9%)	637 (44.5%)	32 (69.5%)	375 (79.4%)
	Civilian service member	150 (44.0%)	60 (37.7%)	236 (39%)	77 (56.6%)	757 (53%)	13 (28.3%)	97 (20.6%)
	Reserves/other	15 (4.4%)	12 (7.5%)	0 (0%)	2 (1.5%)	36 (2.5%)	1 (2.2%)	0 (0%)

* low and middle education presented together as *N* < 6 for some categories.

**Table 2 ijerph-21-00625-t002:** Characteristics of the intervention.

The Intervention Duration	Median	P25	P75
Duration of consultations (minutes) (*N* = 338)	53	47	60
Length of the coaching trajectory (days) (*N* = 255)	196	135	316
Number of consultations (*N* = 341)	6	4	9
**Goals set**	** *N* **	**(%)**	
Lose weight	130	38.1	
Improve diet	85	24.9	
Improve physical activity	119	34.9	
Improve lifestyle	67	19.7	
Improve sleep	49	14.4	
Improve work–life balance	41	12.0	
Improve energy	47	13.8	
Improve mental health	87	25.5	
**Extent to which goals were reached**	** *N* **	**(%)**	
Completely reached goals	68	19.9	
Almost reached the goals	230	67.4	
Did not reach the goals	29	8.5	
No goals were set	14	4.1	

**Table 3 ijerph-21-00625-t003:** Change in health, mental well-being, and sustainable employability in 6 months.

	M0		M1		Difference
	M(SD)	Median	M(SD)	Median	Z
**Health**					
General (1–5)	3.05 (0.68)	3.0	3.33 (0.73)	3.0	−7.338 **
BMI (17.56–40.76)	26.20 (3.70)	25.59	25.75 (3.61)	25.38	−4.820 **
**Mental well-being**					
Work engagement (1–5)	3.48 (0.71)	3.67	3.59 (0.70)	3.67	−3.344 *
Burn-out (1–3.75)	1.77 (0.46)	1.75	1.70 (0.50)	1.75	−2.707 *
**Sustainable employability**					
Recovery after work (1–5)	2.76 (0.73)	2.67	3.01 (0.70)	3.0	−6.889 **
Work–home balance (1–5)	4.17 (0.82)	4.0	4.27 (0.69)	4.0	−2.307 *
Work ability (2–10)	7.47 (1.27)	8.0	7.79 (1.20)	8.0	−4.513 **

* *p* <0.05, ** *p* < 0.001.

## Data Availability

The data supporting the findings of this study are not available due to limitations in sharing data that were collected within the military.

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
