# Peer review of "A Workplace Health Promotion Program for a Predominantly Military Population: Associations with General Health, Mental Well-Being and Sustainable Employability"

_ijerph, 2024, doi:10.3390/ijerph21050625_

Round 1

Reviewer 1 Report

Comments and Suggestions for Authors

I have read the "Health and Readiness: Associations between participation in a workplace health program and health and sustainable employability in a predominantly military population" It is a well-written paper that fully acknowledges the limitations of the study in relation to the abscence of a control group but it also provides in good detail the research  processes followed.

The authors in lines 86-90 explain that the programme run in diifferent period across the the departments with some appearing to have been completed before the beginning of COVID-19 pandemic while other well completed in 2021. Did the authors notice any difference in the delively of the programme or participation in it because of COVID-19?

-Were there any differences in base-level assessments (e.g. mental wellbeing, suatainable improvement) across the participating departments? Due to differences in the types of organisational contexts those individuals belong to, it would be very useful to see if any differences emerged both at pre and post intervention measurements.

- On a final note the authors do not acknowledge enough that such interventions although useful their effects would be always limited as by default they do not address work-factors that can be a source of poor wellbeing at work.

Author Response

Dear reviewer,

Thank you for your time reviewing our paper. Below we provide point-by-point response for your comments on our paper. The adjustments are also highlighted in the main text.

The authors in lines 86-90 explain that the programme run in different period across the departments with some appearing to have been completed before the beginning of COVID-19 pandemic while other were completed in 2021. Did the authors notice any difference in the delivery of the programme or participation in it because of COVID-19?

The program indeed ran in different periods across departments, and during the total period of the program, COVID-19 started in the Netherlands. The program for the Royal Netherlands Army Staff took place from November 2018-November 2020. The program for the Central Staff took place from March 2019-March 2021. The program for the Royal Netherlands Airforce Staff took place from October 2019-April 2021. This indeed means that for some the program was completed before the beginning of COVID-19, and for some the program took place during Covid-19.

On the 12th of March 2020, people in the Netherlands were asked to work from home. There was a shift in the delivery of the program, with the coaching sessions taking place online. When looking at the participation rates, the Army Staff had the highest rate of people participating at baseline (43.3%) and also the highest rate of people completing the 6-month questionnaire (60.9%). This could possibly be explained by the fact that COVID-19 occurred only at the end of the total time period of the program. However, data to confirm this is not available.

A sentence about the influence of COVID-19 has been added in the methods section (line 137-140):

Initially, the meetings took place face-to-face. However, as COVID-19 regulations were initiated in March 2020 and people were asked to work from home, meetings also took place online. Research suggests that online coaching is equally effective as in-person methods’.

Additionally, a sentence has been added to the discussion section, explaining that the results might be explained by the effects of COVID-19 (line 378-382).

A specific confounder that could for example explain the positive effects of the program, is COVID-19. As people were asked to work from home during the pandemic, this might have improved their work-life balance and burn-out complaints as they saved time commuting. However, for others it might have increased burn-out complaints as the schools and daycare centers were also closed.’

Finally, a sentence has been added to the discussion where participation rates are discussed. It has now been added that these rates might also depend on the timeline of the program and the influence of COVID-19 (line 345-346).

‘or due to the time period the program took place and the influence of COVID-19.’

Were there any differences in base-level assessments (e.g. mental wellbeing, sustainable improvement) across the participating departments? Due to differences in the types of organizational contexts those individuals belong to, it would be very useful to see if any differences emerged both at pre and post intervention measurements.

It is indeed useful to see if differences emerged between the different participating departments. Thank you for bringing this to our attention. We ran additional analyses and they showed the there was also a difference across participating departments only for work engagement at baseline. For the other measures, there were no differences across departments at baseline or after 6-months. Additionally, the difference between baseline and the 6-month questionnaire did not depend on military department. We have added the following information to the results section (Line 285-294):

When comparing the different military departments in their scores of health, mental well-being and sustainable employability at baseline (M0) and after 6-months (M1), the one-way ANOVA showed no significant differences across the departments except for the baseline measure of work engagement (F(2)=5.285, P<.05). The Army staff scored significantly higher on baseline work engagement (M=3.61) compared to the Central Staff (M=3.36) and the Airforce Staff (M=3.38).

Additional analyses (data not presented) showed that the 6-month differences presented in Table 3, were not dependent on military department, sex, age, or educational level of participants, indicating that the WPH Program was effective for these different subgroups.

On a final note the authors do not acknowledge enough that such interventions although useful their effects would be always limited as by default they do not address work-factors that can be a source of poor wellbeing at work.

This is an important issue that you address. Thank you for bringing this to our attention. We have added a paragraph to the discussion section (line 362-369).

‘Finally, is important to acknowledge that interventions such as the current WHP Program are limited in their effects, as they do not address work factors that can be a source of poor well-being at work. The interventions focus on how individuals can manage work factors such as workload through healthy lifestyle changes, but they do not focus on decreasing the workload on an organizational level when needed. However, when organizations provide employees with the opportunity to invest in their well-being at work through WHP programs, this gives a signal to employees that their well-being is a priority to the organization.’

Reviewer 2 Report

Comments and Suggestions for Authors

First of all, I would like to thank the authors for the presented results of their investigation on the Netherlands Armed Forces staff. Also, I would like to thank the editor for the opportunity to review this manuscript.

The manuscript “Health and Readiness: Associations between Participation in a Workplace Health Program and Health and Sustainable Employability in a Predominantly Military Population” examines the influence of health program promotion on health and sustainable employability improvements. The authors present an interesting topic that falls within the aims and scope of IJERPH, Special Issue: Exercise and Physical Activity in Health Promotion. Since health program promotions have been studied in various populations, as well as military members, new insights on this topic are always welcome.

Although the manuscript structure is correct and easy to follow, there is room for improvement. First, the title could be more precise in the last part. Secondly, there needs to be more balance between the work objectives and goals and the representation of the entire process that led to the final sample. Either all parts that are not relevant to the objectives and goals of this research should be removed, or the objectives and goals of the paper should be expanded to accommodate the results and discussion. Thirdly, and most importantly, explaining the program itself as much as possible is necessary. In this way, it would be difficult to repeat the research, and the results themselves can be dominantly attributed to the trainer's skills (which is different from the goal of this research).

Good luck

Author Response

Dear reviewer,

Thank you for your time reviewing our paper. Below we provide point-by-point response for your comments on our paper. The adjustments are also highlighted in the main text.

Although the manuscript structure is correct and easy to follow, there is room for improvement. First, the title could be more precise in the last part.

Thank you for taking the time to review our paper. The title has been adjusted (line1-4) to: ‘A workplace health promotion program for a predominantly military population: associations with general health, mental wellbeing and sustainable employability.’

Secondly, there needs to be more balance between the work objectives and goals and the representation of the entire process that led to the final sample. Either all parts that are not relevant to the objectives and goals of this research should be removed, or the objectives and goals of the paper should be expanded to accommodate the results and discussion.

We believe that is it important to provide information on the entire process that led to the final sample, in order to be able to correctly interpret the results. Therefore we have adjusted the objectives of the study, to accommodate the results and discussion (line 83-84).

The objective of this study was to examine associations between participating in the WPH Program and health and sustainable employability improvements over six months and to provide insight into the WHP Program and its strengths and limitations.

Thirdly, and most importantly, explaining the program itself as much as possible is necessary. In this way, it would be difficult to repeat the research, and the results themselves can be dominantly attributed to the trainer's skills (which is different from the goal of this research).

We have included more details on the program itself, in order to provide more clarity. Additionally, we have adjusted the structure of the methods section, to make it easier to find the details about the program. The following information has been added:

Line 109: The program will be described in more detail below.
Line 111-112: Figure 1 has been adjusted.
Line 113: Heading ‘Kick-off’ added.
Line 119-121 & line 123: Information added on baseline questionnaire.
Line 126-130: Information added on the initial intake meeting.
Line 136-146: Information added about the actual coach trajectory.
Line 151-152: Information added on the possibility to do a health check.

Reviewer 3 Report

Comments and Suggestions for Authors

Thank you for the article on the results of the workplace health promotion programme in the Netherlands Armed Forces.

The article is very well written, the results are interesting and the additional information on the NAF is very good in my view.

Nevertheless, I have a few comments that I would like to be taken into account when revising the article.

Firstly, please check the article as a whole in terms of content against the STROBE Guidelines and add any missing information: https://www.equator-network.org/reporting-guidelines/strobe/

This includes naming the tool used for the online survey and the software used for the statistical analyses.

Please also check the spelling of the text again. For example, in line 76 "it is" instead of "It Is".

Line 118: Is the citation incorrect here? Authors cannot be found in the reference list.

Line 193: You mean P instead of Alpha?

Line 336: The bullet point is missing.

Comments on the Quality of English Language

English language is fine. Only minor corrections in spelling are necessary.

Author Response

Dear reviewer,

Thank you for your time reviewing our paper. Below we provide point-by-point response for your comments on our paper. The adjustments are also highlighted in the main text.

Firstly, please check the article as a whole in terms of content against the STROBE Guidelines and add any missing information: https://www.equator-network.org/reporting-guidelines/strobe/

This includes naming the tool used for the online survey and the software used for the statistical analyses.

We checked the article against the STROBE guidelines and we have added the following information:
Line 210-211: Statistical analyses were performed using IBM SPSS Statistics 25.
Line 162-163: The tool ‘LimeSurvey voor Defensie: Versie 1.91+ Build 10746’ was used, the internal survey tool of the NAF.
Line 95-96: Everyone in the specific departments was allowed to participate, there were no eligibility criteria

Additionally, more information about the intervention itself has been added:
Line 109: The program will be described in more detail below.
Line 111-112: Figure 1 has been adjusted.
Line 113: Heading ‘Kick-off’ added.
Line 119-121 & line 123: Information added on baseline questionnaire.
Line 126-130: Information added on the initial intake meeting.
Line 136-146: Information added about the actual coach trajectory.
Line 151-152: Information added on the possibility to do a health check.

Please also check the spelling of the text again. For example, in line 76 "it is" instead of "It Is".

Thank you for bringing this to our attention. The text has been checked, and we have corrected a couple of spelling mistakes. These have been highlighted in the text.

Line 148: Is the citation incorrect here? Authors cannot be found in the reference list.

We have added the citation to the reference list.

Line 219: You mean P instead of Alpha?
This has been adjusted.

Line 411: The bullet point is missing
Bullet point has been added.

Round 2

Reviewer 2 Report

Comments and Suggestions for Authors

Dear Authors,

Thank you for your effort and time. The manuscript is sufficiently improved.

Best wishes.